# Managing Digital Presence in Wineries Practicing Heroic Agriculture: The Cases of Ribeira Sacra and Lanzarote (Spain)

**Elena Cruz-Ruiz** [1], **F. J. Cristòfol** [2] **and Gorka Zamarreño-Aramendia** [3,*]

1 Department of Economics and Business Administration, University Malaga, 29013 Malaga, Spain
2 Department of Communication and Education, Universidad Loyola Andalucía, 41704 Sevilla, Spain; fjcristofol@uloyola.es
3 Department of Economic Theory and Economic History, University Malaga, 29013 Malaga, Spain
* Correspondence: gzama@uma.es; Tel.: +34-687-507-807

**Abstract:** Wine tourism has become an exciting avenue of development for rural wine-producing regions. The channels through which these millenary traditions are transmitted are diverse, and the wineries that practice heroic viticulture can sustainably influence the economic recovery process, especially after the COVID-19 crisis. This paper analyzes the possibilities offered by social media to promote rural territory and wine production in a sustainable way through wine tourism. For this purpose, we have used the case study of the wineries of the Ribeira Sacra appellation of origin and the Canary Islands context in the Lanzarote AO. The methodology used has counted, on the one hand, the existence of web pages of the geographical demarcations mentioned. On the other hand, a study of these wineries' presence on the three main social networking sites, Facebook, Twitter and Instagram, has been carried out. The main focus has been on a content analysis of the social network Instagram, taking into account the terroir's emotional values and tourist attractions. The results show that their presence on the Internet could be higher, as only 55% of the wineries of these appellations of origin have a website. In the case of Instagram, the percentages decrease in Ribeira Sacra and increase in Lanzarote. Finally, it has been possible to trace a model of rural development supported by sustainable tourism, wherein emotional values and transmitted attractions stand out, alongside elements related to the landscape and nature and the wine landscape, tradition and gastronomy.

**Keywords:** heroic viticulture; social media; Ribeira Sacra; Lanzarote; rural development; wine tourism; marketing

## 1. Introduction

Environmental, economic and social sustainability is present and integrated into the management of the territory's resources, and closely linked to the most current wine culture. The tourism options of geographical demarcations (understood as territories, or areas based on specific criteria such as physical features, or cultural differences) with a long winemaking tradition have gained notoriety in Europe and Spain [1–6].

Moreover, the pursuit of sustainable development objectives in wine tourism is one of the factors driving the growth of the sector, based on the enhancement of the value of the territory of appellation wines and their particular link with the terroir [7,8].

The gastronomic and oenological culture is presented as a mixed tourist resource, exerting a unique power to attract wines that possess that quality that links them to the land [9]. Furthermore, it is in this scenario that wineries can contribute positively to gastronomic tourism through wine tourism experiences [10], promoting the place's culture [11].

Tourism that links enogastronomy, as mentioned above, with the diversification of farms seeking local development [12] enhances the prospects of rural areas [13–15].

In addition, wine tourism represents an exciting source of income in which many countries enjoy a privileged position, having resources that generate better options for

the future in the framework of wine tourism (Figure 1). In addition to this, the growing production of organic wines allows better final prices to be obtained in the market [16].

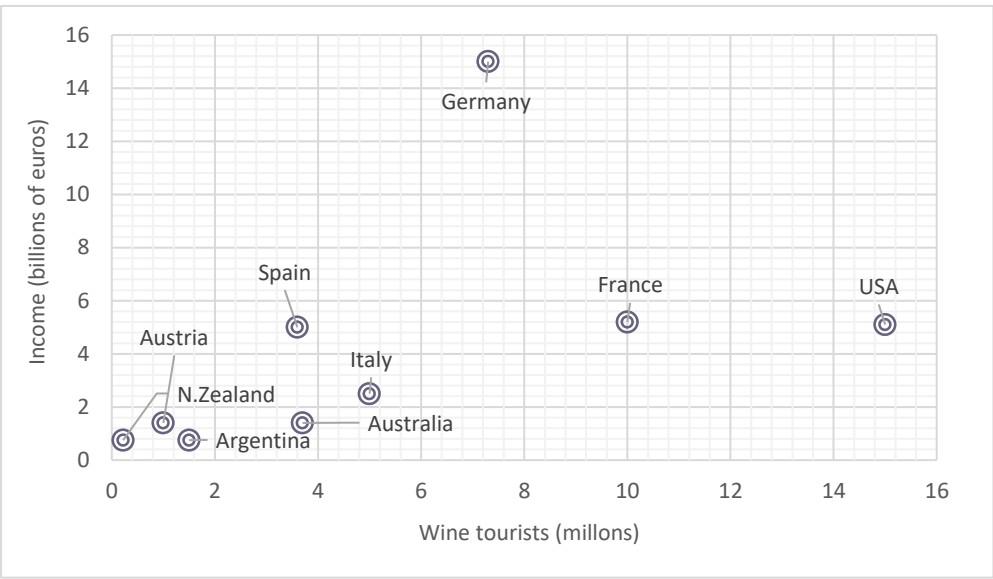

**Figure 1.** Wine tourists and wine tourism revenues in main countries (2019). Source: Own elaboration with data from Vins du Monde (Accueil (2021) Vins du monde. Available at: https://www.vinsdumonde. fr/ (accessed on 20 March 2023)).

The effects of the COVID-19 crisis are still present, with repercussions on consumption and wine tourism in Spain [17] harming wine exports and the international tourism market [18,19]. The data source Statista shows that Spain was the country most affected by the pandemic, losing up to 91% of its wine tourists, a figure similar to that of Italy, at 86%, and Portugal, at 85%, while France saw its numbers reduced by 77%. On the other side of the Atlantic, the USA experienced a 75% drop. The best performers were Austria and Germany, with 66% and 44% declines, respectively.

However, although business will continue to recover, proper management of wine tourism can improve this situation through wineries' promotional campaigns [20]. Moreover, at this point, social networks will reinforce their role as crucial tools for organizations' communication, both in the Spanish wine sector and in other markets wherein wine tourism is important [21]. New technologies and social media are a perfect vehicle to energize the sector, taking into consideration the place's cultural and heritage values. This is where agriculture and vineyards provide new wealth-generating functions [22].

In this context, the territory becomes an element of communication [23], where heritage resources are increasingly present in the digital promotional media that wineries and institutions make available to consumers [24].

The purpose of this research is to analyze the management of the information offered on websites and in the most relevant social networks of wineries that practice mountain viticulture, providing, in this case, a boost to the attractions of the environment, wherein the landscape represents an added value to wine tourism.

Specifically, heroic viticulture is the process of planting and harvesting grapes of an ancient tradition. Manual procedures harvest the fruit on land with slopes above 30% and an altitude above 500 m [25]. See Figures 2 and 3, representing the Ribeira Sacra and Lanzarote AOs, which show these singularities.

The chosen areas, members of CERVIM (Center for Research, Study, Safeguarding, Coordination and Valorization of Mountain Agriculture), are hallmarks of the value of wine landscapes and have already been analyzed in the case of Ribeira Sacra by De uña- Alvarez and Villariño Perez, [26] and López Varela, [27] and in the case of Lanzarote by Hernández et al., [28], by Zottele, and González Santana, [29]. The aforementioned studies have not

considered the management and marketing of the social media used, instead focusing their interest on the observation of the landscape resource as an enhancer of wine tourism in these and other regions that practice heroic viticulture, such as Italy (Zottele et al., 2000) or Okanagan Valley (Canada) [30], among others.

The Ribeira Sacra AO is located in Galicia (Spain) and was recognized in 1996. However, it has a long history; in the year 876, the donation of vineyards to the monastery of Santa Cristina de Sil to fulfil Christian devotions is documented [31]. The Ribeira Sacra represents a unique territory in which it is difficult to dissociate the natural landscape from the cultural heritage [32], (Figure 2).

In the case of the Canary Islands, the winemaking tradition dates back to ancient times. The Appellation of Origin Wines of Lanzarote was created in 1993. One of its distinctive features is that it covers the entire surface of the island, with a particular interest in the unique landscape of the area of La Geria, as this is perhaps where the winegrower must make a more significant effort to overcome the barriers imposed by nature (Figure 3).

The research developed raises the following research questions:

RQ[1] What does the digital presence of Ribeira Sacra wineries and Lanzarote wineries contribute to the development of the product and the rural environment?

RQ[2] Are there values that bring us closer to a communication model of wineries through their digital presence on Instagram?

RQ[3] Is the management of digital media used to promote the terroir adequate?

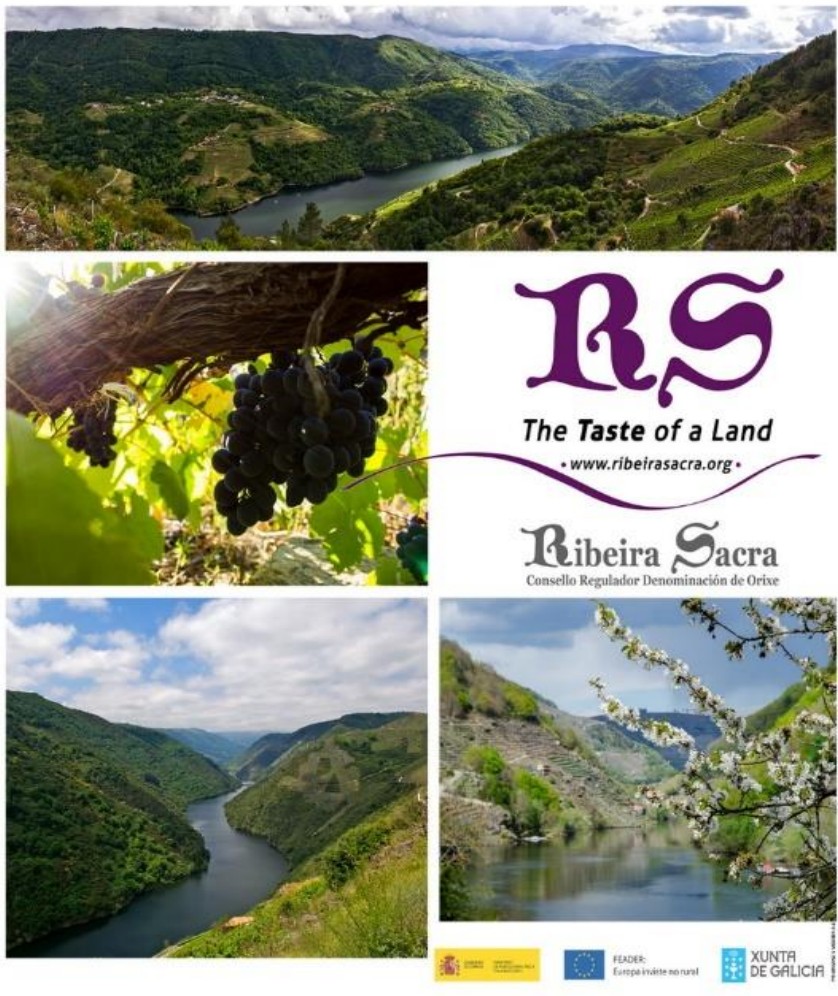

**Figure 2.** Scenarios of heroic viticulture in Ribeira Sacra (Galicia, Spain). Source: Regulatory Council of the Appellation of Origin Ribeira Sacra [33].

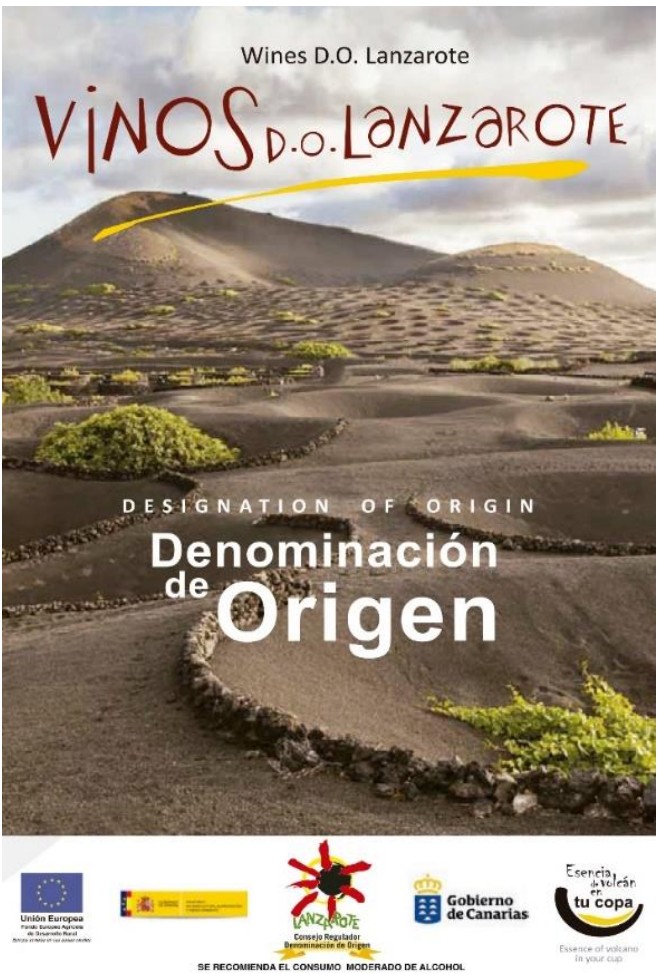

**Figure 3.** Scenarios of heroic viticulture in the Lanzarote AO, (Islas Canaria: España). Source: Regulatory Council of the Appellation of Origin, Lanzarote [34].

The research structure starts with an introductory chapter reflecting the interest in heroic viticulture in two wine tourism geographical areas in Spain. Section 2 presents a literature review focused on the importance of terroir in the framework of the extraordinary technique used and the interest of social media for the promotion of wineries in the analyzed territories. Section 3 gathers the methodological aspects that have been carried out through a systemic analysis of the contents that the wineries deposit in their networks, especially through observation of Instagram. The strong emotional component on which Instagram—and concomitantly, visual storytelling—is built allows companies to be successful on the platform [15,30]. Instagram's focus has always been on visual content. The results of the study are shared in Section 4, in which it is found that the wineries' digital presence is generally low. In the case of Instagram, values related to tradition and attractions such as landscape and nature and wine landscape, tradition and gastronomy are mainly transmitted. The discussion and conclusions close the research.

## 2. State of the Art

In Spain, the relevance of wine tourism began to be present in the academic literature in the late twentieth century, significantly when official wine routes were consolidated [35,36].

These routes satisfy increasingly demanding tourists who are looking for an experience that wine routes provide, thereby encouraging travel [37]; wines may even be tourists' primary motivation [38,39]. Academic literature has analyzed the various impacts of the communication and marketing elements of wine tourism. One of its focuses is the use of social networks [10,40]; some examples of case studies are that of Belias et al. [41] for Crete,

Carvalho et al. [42] for the Portuguese region of Bairrada, or Canovi and Pucciarelli [43] for the Italian case of the Langhe region.

The proposed research undertakes a comparative study of two Spanish areas of production that fall within the territories of heroic agriculture and which possess a series of characteristics that differentiate them from other wine-growing areas.

### 2.1. The Wine Landscape and Terroir

The term terroir is one of the most controversial in the world of wine, and one of the most used and least understood [44]. A definition proposed by the International Organisation of Vine and Wine (IOV) refers to the originality of the wine produced in a terroir. In this sense, terroir is a concept relating to an area in which the collective knowledge of the interactions between the identifiable physical and biological environment and the viticultural practices applied is developed, providing distinctive characteristics to the products from this area. Terroir includes specific characteristics of soil, topography, climate, landscape and biodiversity [45].

A classic definition is that of Seguin [46], stating that terroir recognizes an interactive ecosystem in a given place, including climate, soil and vine. The terroir is a spatial and ecological concept that connects the actors, histories, social organizations, activities, and agricultural practices [47]. For most European anthropologists, terroir is expressed through the product to which it confers its originality in the sense of a specific product [48]. It provides a connection to nature and a specific locality in an increasingly globalized world [49]. For many, this sense of place—the sense of origin and authenticity—embodies the ulterior meaning of wine quality. In contrast, for others, it is a marketing tool based on the clichés of the territory wherein the wine is produced [3].

In addition, the different systems for certifying wine quality based on Protected Appellation of Origin (PAO) and Protected Geographical Indications (PGI) take for granted that the land on which the grapes are grown represents unique and recognizable qualities specific to the region. These products cannot be reproduced anywhere else in the world. In this sense, terroir describes the total natural environment of particular vine-growing sites [50], which includes climate, as measured by temperature and rainfall; solar energy (or insolation) received per unit of land area; relief (topography or geomorphology), which includes altitude, slope and aspect; geology and pedology. These conditions determine the essential physical and chemical characteristics of the different soil types, the terroir's hydrology and the soil–water relationships in question [51].

Terroirs indicate a direct relationship between the characteristics of an agricultural product (quality, flavor, style) and its geographical origin, which can have a significant impact on these characteristics and is therefore recognized as an essential factor in wine quality and style, especially in European vineyards [52].

It is essential to consider the interactions between the constituent factors. Thus, the elements of the natural environment (or of the physical environment) are present and interact with the vineyards, influencing vine growth, grape ripening and the sensory attributes of the wine, hence their importance [51].

On many occasions, it is not easy to dissociate the natural landscape from the cultural heritage, since the anthropic footprint on the territory can be seen in the heritage elements that have marked the terroir over time [53].

The terroir should also be understood as a socio-emotional construction derived from the traditions, historical stories and symbology that connect it with the past. Thus, interest in preserving the traditions of a particular territory, including its cultural heritage, reputation and tradition, is critical to acquiring a competitive advantage in the face of an extensive market [54].

### 2.2. Wineries and Digital Culture

Tradition and innovation are present in the wine sector and bring competitive advantages in many wine regions [55]. A step forward in the search for modernity and

competitiveness is the promotion of oneself through the use of new technologies and the adoption of social networks as new forms of digital communication which are faster and more direct [56,57].

Digital culture comprises a set of customs and forms of social interaction that are carried out thanks to the resources offered by digital technology [58]. The use of social networks has imposed itself in recent years as part of the communication plans of the wine tourism industry, changing the marketing dynamics of companies [59] and making it possible to reach potential customers. Moreover, the use of social networks is positively related to the purchase of wine online [60].

In addition, the wine industry in a developing country can enhance its competitive advantage by correctly using its social networks [61]. Communication can be fundamental to identifying the uniqueness of the territories wherein heroic viticulture is practised [62]. It primarily represents an additional opportunity that consumers can perceive through elements that identify these distinctive elements in a timely manner [63], allowing producers to conduct campaigns that reveal these differentiating characteristics [64].

Currently, wine tourists resort to informal and external channels to connect with the wineries themselves, where the information generated by previous visitors has acquired great importance [65]. Thus, the search for information should be controlled by wineries in terms of brand and experience [66] and supported by the use of websites, as they are generally the first point of contact of customers with wineries, their wines and activities [67].

Jeong, Holland, Jun and Gibson [68] point out that any tourist destination must have a web domain so that the correct use of this communication channel positively impacts the intention to visit [69]. Websites are a magnificent showcase, offering multiple possibilities beyond mere information; events, wine sales and valuable information for visitors go hand in hand. It should not be forgotten that there should be links to social networks to induce customer participation and interaction with the business ecosystem [67].

In this way, social networks have become a fundamental means for sharing audio-visual content from wineries and improving their ability to interact directly with customers, as there is an increase in network conversation [70]. The result facilitates the generation of a brand community [71].

Decision making is closely related to consumer interaction, and participation results in more intense and greater possibilities for content creation [72]. These interactions have the advantage of increasing the degree of trust in the information offered by wineries, since visitors are more inclined to take into account the opinions of acquaintances or stakeholders in the world of wine [65].

Social networks are of great importance for recalling experiences to the extent that they build loyalty among customers who may repeat their visit or recommend it to friends and family [73]. Value generation is one of the most important aspects to be taken into account in social networks; it should transcend the mere promotional element, connecting the winery with its followers [74].

Wine tourists using the social networks of Instagram and Facebook, among others, allow winery companies better promotion to the extent that they explore the attitudes, values and motivations of wine lovers and wine consumers on the Web [24,75].

## 3. Methodology

The study area comprises two geographically differentiated Spanish areas within the territories of heroic viticulture.

Ribeira Sacra is located in the northwest of Spain, specifically in the province of Lugo, in the autonomous community of Galicia. The name "Ribeira Sacra" means "sacred riverbank," It refers to the banks of the rivers Sil and Miño which run through the area. Steep river valleys, terraced vineyards, and rugged mountains characterize the landscape of Ribeira Sacra. The region has a mild climate with abundant rainfall, making it an ideal place for cultivating grapes, chestnuts, and other crops. The area is known for its rich cultural heritage, including numerous Romanesque churches, monasteries, and castles.

Ribeira Sacra is also well known for its wine production, particularly red wines made from the Mencía grape. The steeply terraced vineyards are a distinctive feature of the region, and many wineries offer tours and tastings to visitors.

Lanzarote is one of the Canary Islands located off the coast of Morocco in the Atlantic Ocean. The island is known for its unique volcanic landscapes and a warm, subtropical climate tempered by cool sea breezes. The Lanzarote wine terroir is characterized by its unique volcanic soils and climate. Over the centuries, the island's volcanic eruptions have left a distinctive layer of black volcanic ash and pumice over much of the island, creating fertile soil rich in minerals and nutrients. The vineyards on Lanzarote are typically planted in shallow, and cone-shaped depressions dug into the volcanic ash, known as "Hoyos." The walls of these Hoyos are made from the same volcanic ash, creating a natural windbreak that protects the vines from the firm, dry winds that blow across the island. The combination of the volcanic soil, the unique growing conditions, and the variety of grape, Malvasia Volcanica, grown on the island gives Lanzarote wines a distinctive flavour profile.

The following research has analyzed the digital presence of the wineries of the Ribeira Sacra and Lanzarote AOs, including the existence of websites, as well as profiles on the most relevant social media sites, Facebook, Instagram and Twitter.

The study focuses particularly on publications on their Instagram profiles. The method of coding by categories and classifying the audio-visual documents related to the product they promote has been used, observing the frequency of publication and the interaction they generate in the aforementioned social network. The impacts generated by publications on Instagram seek to influence users to engage with the wine tourism experience. For this reason, the following two aspects are studied: tourist attractions and the emotional values that appear in the accounts of this social network.

When this study is limited to the Instagram social network, it is in order to show the social network that gives audio-visual content the most value. Instagram is also the social network used most on a daily basis by Spanish users, according to the IAB Spain Social Network Study [76]. Instagram is used by 67% of users several times a day or every day. The main functions of the network are to show audio-visual content which is increasingly linked to video but always features the crucial element of the image. The average usage time of Instagram is one hour and thirteen minutes, and in 2022, compared to 2021, the frequency of visits increased. According to the same source, the users' perception of Instagram is that of a specific network of photos and fun.

In order to answer the research questions, a content analysis has been carried out; this is understood as the application of a method of study and analysis of communication in a systematic, objective and quantitative way, intending to measure certain variables [77].

First, the research should answer whether the wineries in the territories studied have a digital presence. Official sources were used to recognize the wine producers registered in the areas under study. Digital presence was understood as the existence of an official website or presence on one or all of the leading social networks, Facebook, Instagram and Twitter.

Secondly, a systematic qualitative analysis of the data obtained was carried out using NVivo software. This software stands out as a tool capable of automating non-numerical data such as interviews, surveys and textual content that helps researchers [78] in the treatment of qualitative data by allowing the elaboration of models through intersection matrices [79].

The photography work environment enabled by NVivo software is a work environment that handles different representational languages, making it simple to access data, and that allows us to organize photography and the meanings that its author gives it in the processes of photo elicitation. The program has tools for the categorization of material. An important aspect to take into account to ensure the quality of the analysis is that each code has a description section, where qualifying the information related to it promotes the explanation of the material collected there so that researchers proceed in analysis with a network of shared meanings, thereby guaranteeing the quality of the process. Under this premise, Table 1 was elaborated, wherein the 18 tourist attractions considered relevant by Huertas ([80]) are collected, along with 18 other emotional values).

**Table 1.** Content analysis sheet.

| Tourist Attractions | Emotional Values |
|---|---|
| Landscape and nature | Romanticism |
| Wine landscape | Charm |
| Intangible heritage | Friendship |
| Folk | Escapism |
| Urban landscape | Modernity |
| Urban leisure | Cosmopolitanism |
| Nightlife | Responsibility |
| Shopping | Authenticity |
| Sun and beach | Safety |
| Weather | Joy |
| Gastronomy | Youth |
| Ecology | Dynamism |
| Hospitality | Relaxation |
| Luxury | Tradition |
| Sports as spectacle | Quality of life |
| Sports to practice | Sophistication |
| Plans | Diversity |
| Others | Others |

Source: Own elaboration by authors, based on Huertas [80].

The content analysis focused on the websites of the available wineries, using the FireShot software to capture the covers of all the websites of the Ribeira Sacra and Lanzarote wineries. Subsequently, the emotional values and tourist attractions transmitted by these pages were analyzed (Appendix A), assigning an element in each category per page analyzed according to the proposal of Huertas [80].

Finally, concerning the analysis on Instagram, the starting point was the collection of screenshots of the accounts of the wineries of Ribeira Sacra and Lanzarote active on this social network. These screenshots were taken using the FireShot software between March and April 2022, considering that the pruning period had already ended in both territories. The screenshots include nine graphic posts from each account, since that is the number of images shown on the Instagram front page.

Three tourist attractions and three emotional values have been given to each screenshot, so a complete picture is achieved through a more thorough analysis.

Each of the tourist attractions and emotional values is defined below in Tables 2 and 3. The definitions are adapted to the presence of these concepts in images, not in text or other elements.

**Table 2.** Definition of tourist attractions.

| Tourist Attractions | Definition |
|---|---|
| Landscape and nature | Presence of landscapes or natural elements |
| Wine landscape | Presence of vineyards, vines, plantations or other elements related to viticulture. |
| Intangible heritage | Non-physical heritage elements |
| Folk | Folkloric manifestations of people |
| Urban landscape | Presence of urban elements (buildings, streets, furniture, etc) |
| Urban leisure | City-related leisure: city bars, museums, etc. |
| Nightlife | Nightlife-related leisure: pubs, clubs, discotheques, etc. |
| Shopping | Presence of elements related to shopping (shopping malls, stores, etc.) |
| Sun and beach | Presence of elements related to sun and beach tourism (sand, beach, beach bars, etc.) |
| Weather | Elements related to the weather (sun, clouds, rain, etc.) |
| Gastronomy | Presence of food or wine dishes |
| Ecology | Elements of environmental care |
| Hospitality | Attitudes that evoke closeness and good treatment |
| Luxury | Appearance of luxury elements: brands, boats, private jets, etc. |
| Sports as spectacle | Celebration of sporting events: soccer, sailing, basketball, etc. to be watched |
| Sports to practice | Celebration of sporting events to be practiced: golf, regattas, etc. |
| Plans | Proposal of leisure plans |
| Others | Everything not included in the above |

Source: Own elaboration by authors, based on Huertas [80].

**Table 3.** Definition of emotional values.

| Emotional Values | Definition |
|---|---|
| Romanticism | Evocation of romantic attitudes in a couple |
| Charm | Elements that highlight positive feelings |
| Friendship | Attitudes that evoke the feeling of friendship |
| Escapism | Absence of noise in the images, cleanliness and tranquillity |
| Modernity | Presence of modern elements such as buildings, new technologies, etc. |
| Cosmopolitanism | Presence of different cultural elements or ethnicities |
| Responsibility | Values related to care and attention |
| Authenticity | Presence of values consistent with the reality of the territory |
| Safety | Absence of risk |
| Joy | Existence of elements that show feelings, degrees or gestures and acts that express joy |
| Youth | Presence of elements highlighting actions related to youth |
| Dynamism | Evocation of values related to the activity |
| Relaxation | Appearance of elements that present mental and physical distraction or rest |
| Tradition | Features of a region's identity |
| Quality of life | Presence of elements that emphasize hedonism |
| Sophistication | Elements that highlight elegant and refined values |
| Diversity | Diversity of sexes, genders, races or religions |
| Others | Everything not included in the above |

Source: Own elaboration by authors, based on Huertas [80].

## 4. Results

Wine with Appellation of Origin is the culture and heritage of each region, a resource linked to numerous territories in Spain, where wine and culinary traditions come together in a perfect marriage to attract visitors and enhance the value of the territories.

One of the points of view Getz pointed out concerning wine tourism was the study of the forms of sale and promotion of the business, taking into account the socioeconomic development that can take place in the region as a result of the relationship between tourism and wine (Getz, 2000).

The cases analyzed in the two large areas characterized by heroic viticulture have a significant presence of small family wineries. Social media can be an interesting tool in these areas, as use of social media in many cases is a form of consumer outreach, as pointed out by Obermayer et al. [81], because these technological resources are particularly appropriate among wine consumers, as word of mouth is an essential driver of wine sales [82].

### 4.1. Internet Presence

The analysis of the official sources of the appellations of origin studied yielded the following data, which can be seen in Table 4, wherein the presence of a website exceeds 50% in both cases.

**Table 4.** Wineries of the DD.O. Ribeira Sacra and Lanzarote with a website.

| Territory | Total Number of Wineries | With Website | % Website |
|---|---|---|---|
| Ribeira Sacra | 96 | 50 | 52.1% |
| Lanzarote | 21 | 12 | 57.1% |

Source: Own elaboration.

The presence of the wineries on social networks is presented in Table 5, differentiating the territory and each social network's presence in absolute number and percentage.

**Table 5.** Presence of social media in wineries of the Ribeira Sacra and Lanzarote AOs.

| Territory | Total | Facebook | % Facebook | Twitter | % Twitter | Instagram | % Instagram |
|---|---|---|---|---|---|---|---|
| Ribeira Sacra | 96 | 43 | 44.8% | 14 | 14.6% | 22 | 22.9% |
| Lanzarote | 21 | 16 | 76.2% | 5 | 23.8% | 14 | 66.7% |

Source: Own elaboration.

The case that concerns the present research focuses on the presence of wineries on Instagram. In this regard, of the 96 registered in the Ribeira Sacra Designation of Origin, 22.9%, i.e., 22 wineries, have a presence on this social network; on the other hand, in the Lanzarote territory, 14 of the 21 wineries, 66.7%, use this social network. Figures 4 and 5 shows an example of the use of Instagram by two wineries.

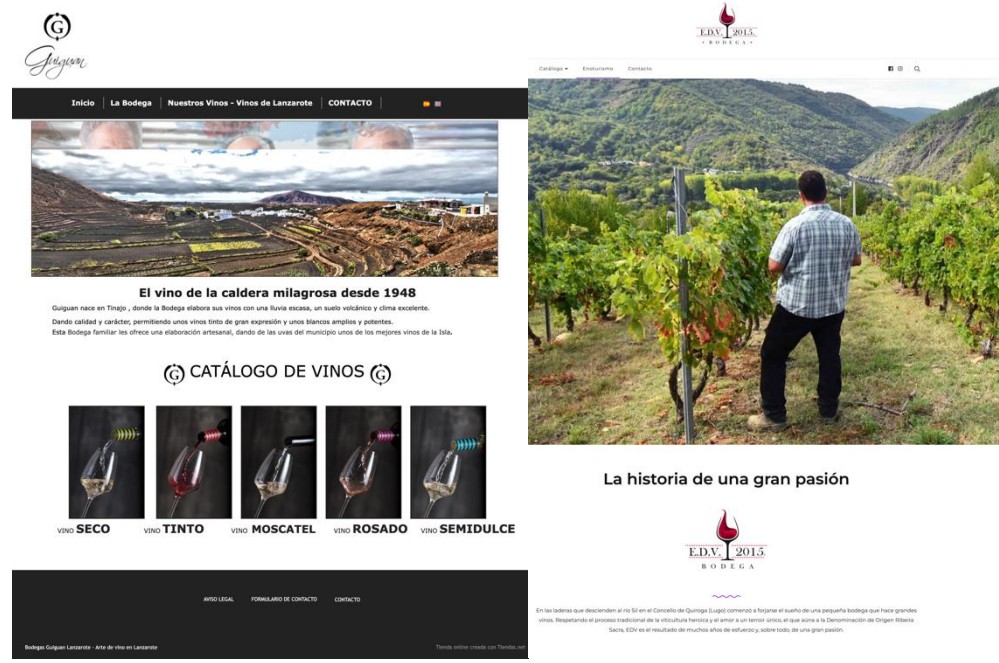

**Figure 4.** Example of websites in the Lanzarote and Ribeira Sacra AOs. Source: Wineries' websites.

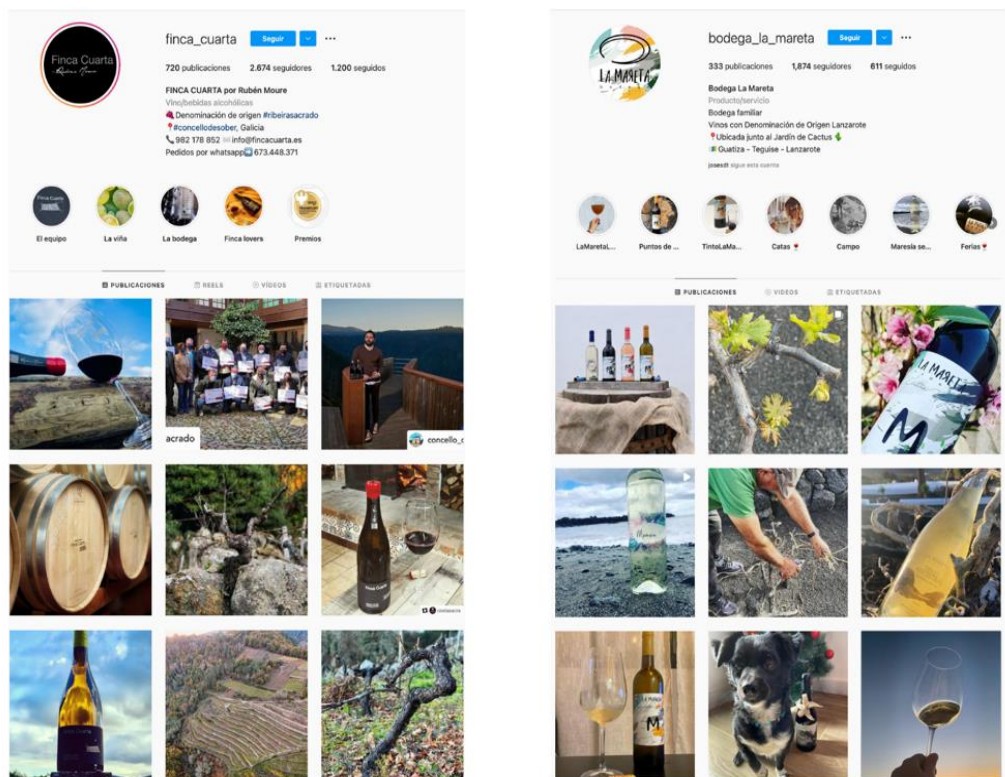

**Figure 5.** Example of Instagram use by wineries from the Ribeira Sacra and Lanzarote AOs. Source: Own elaboration based on publications made by the wineries studied on Instagram.

4.1.1. Ribeira Sacra AO

Of the 50 wineries with a functional website in Ribeira Sacra, 47 have active URLs, i.e., they allow visitors to access the site without displaying an error message when entering the address published by the Appellation of Origin. As shown in Table 6, of these pages, 36 present the wine landscape as a tourist attraction, and 34 focus on tradition as the primary emotional value.

**Table 6.** Touristic attractiveness and emotional value in the websites of the Ribeira Sacra AO.

| Territory | Active Websites | Touristic Attractiveness | Emotional Value |
|---|---|---|---|
| Ribeira Sacra | 47 | Wine Landscape 36 (76.7%) | Tradition 34 (72.4%) |

Surce: Own elaboration.

As shown in Figure 6, the Instagram accounts of the 22 wineries with a presence in this network show two tourist attractions in their publications; some 20 accounts show landscape and nature, and another 19 show the wine landscape. Other categories include gastronomy, with nine profiles showing these attractions, popular culture, with six, and intangible heritage and ecology, with five.

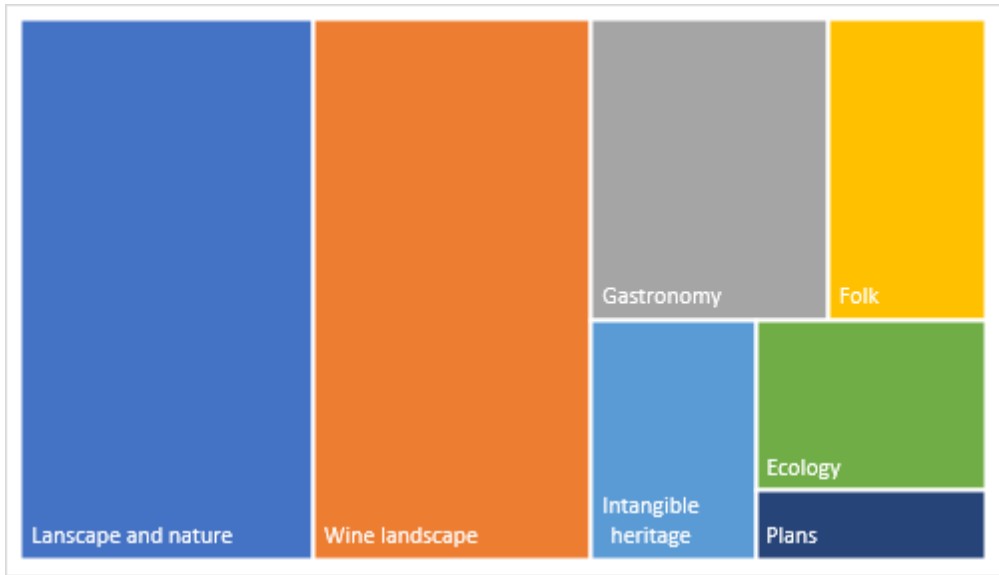

**Figure 6.** Tourist attractions of the Ribeira Sacra AO. Source: Own elaboration.

Figure 7 shows two profiles of wineries in Ribeira Sacra (Vía Romana and Finca Cuarta) wherein the prominence of the wine landscape and the landscape and nature can be seen.

Figure 8 shows the analysis of the wineries of the Ribeira Sacra AO and the transmission of emotional values through their Instagram profiles. Tradition appears as the dominant category, with 17 profiles that present this value. Tranquillity, with 13, relaxation, with nine, or friendship, with eight, are also values that are implied through the profiles.

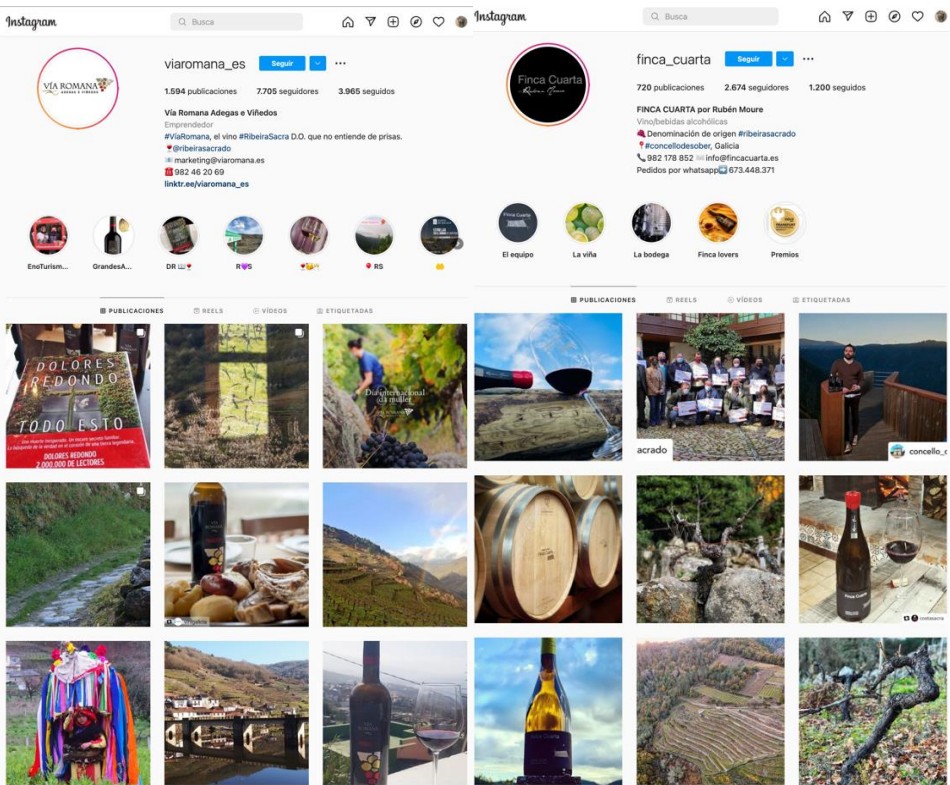

**Figure 7.** Instagram profiles of the Ribeira Sacra AO. Source: Own elaboration.

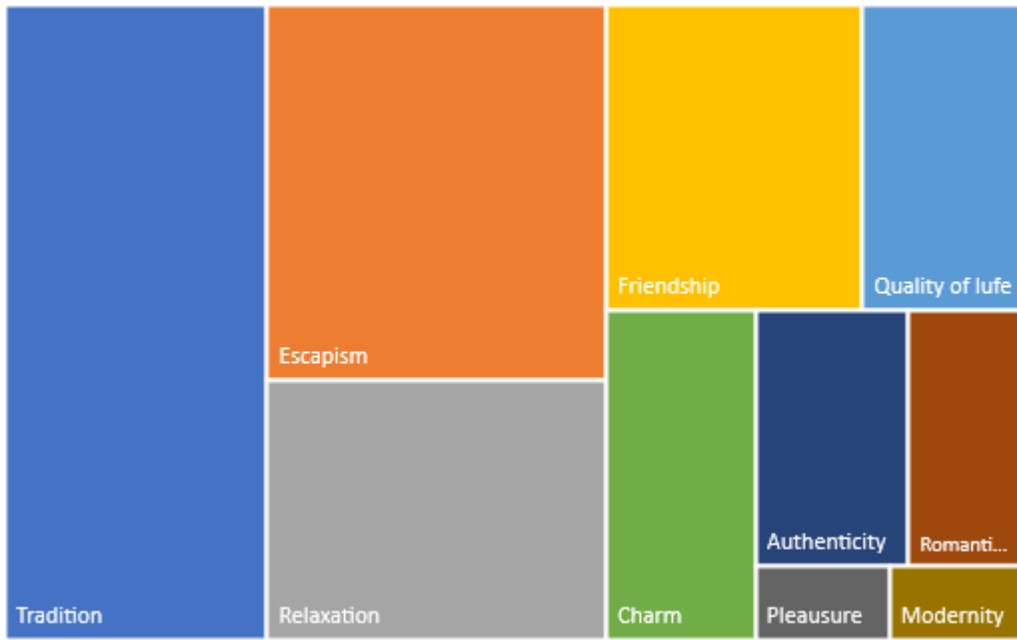

**Figure 8.** Emotional values of Ribeira Sacra wineries' Instagram profiles. Source: Own Elaboration.

4.1.2. Lanzarote AO

Table 7 shows the analysis of the dynamic web pages of the Lanzarote AO. In this sense, the 12 pages referenced in the official source are active and in use. Thus, two out of three project the wine landscape of the area as the main tourist attraction, while 75% transmit the emotional value of tradition.

**Table 7.** Wineries of the Ribeira Sacra and Lanzarote AOs with a website.

| Territory | Wineries | Website | % Web |
|---|---|---|---|
| Lanzarote | 12 | Wine landscape, 8 (66.6%) | Tradition, 9 (75%) |

Source: Own elaboration.

Figure 9, which corresponds to the Lanzarote territory, shows the tourist attractions transmitted by the Instagram profiles of the wineries present in that network according to the systematic content analysis sheet adapted from Huertas [80]. In this case, the wine landscape, with a dozen profiles, the item landscape and nature, with ten, and gastronomy, with nine, are the three most repeated concepts. Ecology also appears on four occasions, urban leisure on three, and popular culture and hospitality on two.

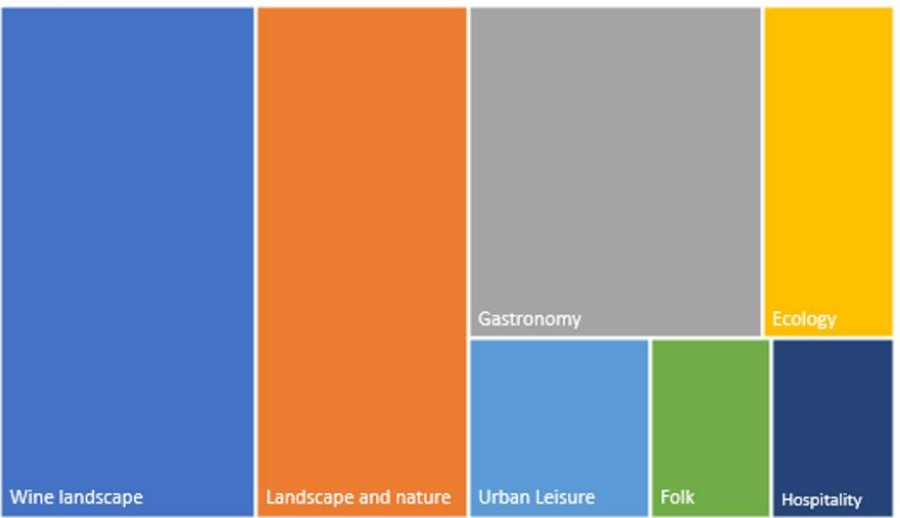

**Figure 9.** Tourist attractions of the Lanzarote AO. Source: Own elaboration.

The emotional values that are transmitted through the Instagram accounts of the wineries in the territory of Lanzarote are mostly tradition and escape, with eight repetitions, as shown in Figure 10. This is followed by relaxation, with seven; charm, with six; quality of life, with five; friendship, with four, and to a lesser extent, authenticity, with two, and romanticism and modernity with one.

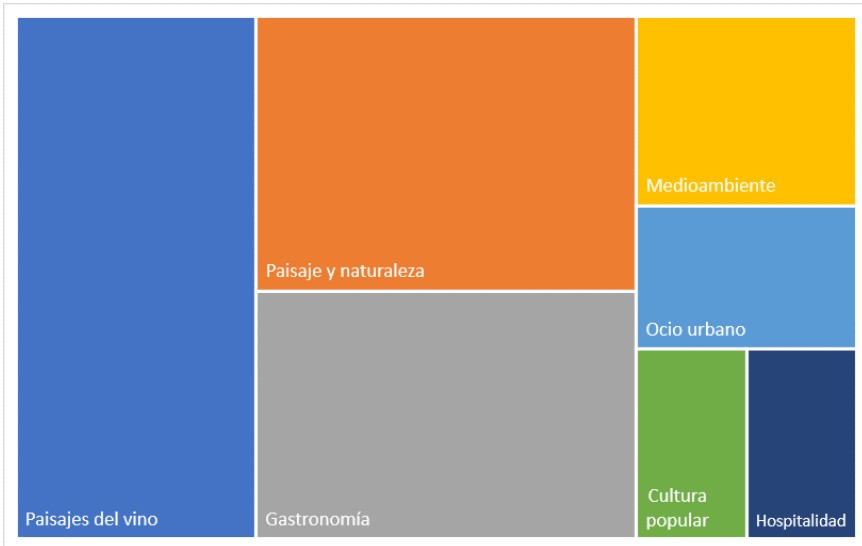

**Figure 10.** Emotional values of the Lanzarote AO. Source: Own elaboration.

Figure 11 shows two profiles of the Lanzarote DO (Bodega La Mareta and Vinos de la Geria). Both show traditional, manual pruning or escapism within relaxing landscapes.

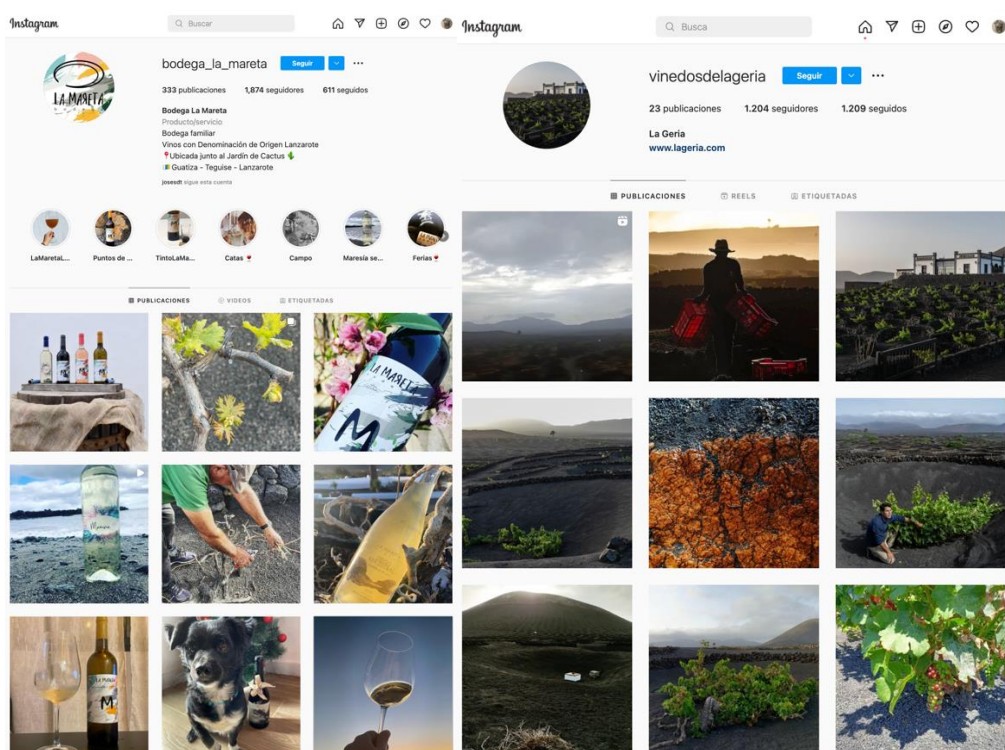

**Figure 11.** Instagram profiles of the Lanzarote AO. Source: Own elaboration.

## 5. Discussion and Conclusions

Wine has gone from being a product of millenary tradition in consumption to favouring the establishment of a development model for rural territories; this requires policies and actions that seek a change in the productive structure to pursue environmental sustainability, as well as the enhancement of the heritage and culture of the villages (Robinson, 2021). This is why the digital presence of wineries in the territories studied mainly focuses on transmitting values such as tradition or tourist attractions related to the terroir.

In studies of the digital presence of brands in the tourism field, it is observed that, in the period between 2016 and 2022, there has been a considerable change The arguments maintained in the study by Huertas [80] have changed. Currently, brands focus more on intangible values such as tradition and gastronomy. In the conclusions of that study, it was stated that the messages of the Spanish tourist brands featured a more significant presence of tangible attributes; in the present study, it is observed that the trend is very different.

The new opportunities derived from the enhancement of traditions, through the options offered by the territories listed by UNESCO [83], indicate that different territories with similar characteristics, such as heroic viticulture, could find synergies with these peculiarities as a tourist resource. That is, highlighting the landscape, the wine and nature itself may help to find a clear point of transversal communication of territorial values, as in the cases of Ribeira Sacra and Lanzarote.

Wine tourism responds to a hedonistic impulse, to the need to know and experience the wines [2] and the place in which they are made [84]; therefore, visits to production areas and wineries are the axes of this type of tourism. The opportunity for wineries is evident, even though some studies indicate how spontaneous wine tourists are when choosing their destinations [85]. The prior information they can find is key to planning their visits [65,86,87], so digital presence becomes mandatory for wineries that aspire to participate in wine tourism routes.

In answering the first research question, it can be concluded that the digital presence of the wineries on the internet in both territories is meagre, since barely 55% of the total have a website. However, in both cases, the tourism values highlighted through their websites are focused on the wine landscape. The relevance of landscapes in both appellations is a reality, and winery managers must be aware that this tourist attraction is an exciting element to highlight. On the other hand, tradition as an emotional value occupies first place in both cases, highlighting the importance of culture in the territories analyzed.

Regarding social networks, there is a significant percentage difference in the presence of wineries on Instagram. In the case of Ribeira Sacra, barely 22.9% of the 96 wine producers have an account on that social network; in the case of Lanzarote, the case is radically opposite, since two thirds of wineries, a total of 14, have an account on that network. In Lanzarote, 12 of the 20 wineries have a website, so their presence on the internet is varied. This is why digital media are essential to publicizing rural values and disseminating sustainable practices as a differentiating value among potential visitors.

In response to the second research question, the research shows that the most popular tourist attractions, which form the fundamental pillar of its tourist model, are the wine landscape and the landscape and nature. In both cases studied, these are the most repeated concepts, reinforcing the value of the heroic viticulture practiced. Moreover, in both cases, gastronomy is an element to be considered, reflecting the close relationship it maintains with wine.

These terroir values have relevance for producers, highlighting that these values and attractions are critical to strengthening the image of wines made from the technique of heroic viticulture, helping to maintain the story of belonging to the territory.

Tradition, understood as the use of traditional techniques such as fining, oak barrels, and traditional grape species, is a significant value that is more present in Instagram accounts. These regions are both mainly traditional territories. In addition, the values of escape and relaxation associated with the territory also present a clear example of the type of destination studied.

On the other hand, heroic viticulture itself is outside the parameters of industrial wine production and is rooted in tradition. This point is vital to understanding why wineries are committed to presenting this in their publications. Likewise, areas such as Lanzarote or Ribeira Sacra, which have a recognized gastronomic quality, also include these values and attractions in their social media strategy.

Finally, in response to the third research question regarding whether the management of the digital media used is adequate, from the point of view of the wineries' business management, there is room for improvement in specific elements.

Rural tourism, seen as sustainable tourism, has several objectives, so the work of the agents of the territory, especially wineries, is fundamental. The population must be aware of the opportunities of wine tourism for the economy and the rural environment, while working to protect the environment and heritage values that ultimately contribute to improving the community's standard of living.

This task could be made more accessible if interconnections were created between the Spanish AOs with heroic viticulture (and later with the rest of Europe), with the aim of generating standard tourism products and collaboration based on open innovation. In this sense, strengthening the concept of heroic viticulture related to wine quality and its close link with the terroir highlights all producers' fundamental and shared values and attractions.

The economy of rural territories requires adequate transmission of the character of the terroir, so their digital presence is a primary tool to meet these objectives.

**Author Contributions:** Conceptualization, E.C.-R. and G.Z.-A.; methodology, F.J.C. and G.Z.-A.; software, F.J.C.; validation, E.C.-R., F.J.C. and G.Z.-A.; formal analysis, E.C.-R. and F.J.C.; investigation, F.J.C. and E.C.-R.; resources, G.Z.-A.; data curation, E.C.-R. and F.J.C.; writing—original draft preparation, F.J.C. and G.Z.-A.; writing—review and editing, E.C.-R.; visualization, G.Z.-A. and F.J.C. All authors have read and agreed to the published version of the manuscript.

**Funding:** This research received no external funding.

**Acknowledgments:** The authors would like to acknowledge the help of the wineries and the Appellation of Origin of Ribeira Sacra and Lanzarote. Special thanks to the University of Málaga.

**Conflicts of Interest:** The authors declare no conflict of interest.

## Appendix A

Wineries Websites
http://adegavella.com/
http://bodegagullufre.com/
http://www.adegaalgueira.com/
http://www.adegacabodomundo.es/
http://www.adegaslareu.com/
http://www.bodegaribada.com/
http://www.diegodelemos.es/
http://www.dominiodobibei.com/
http://www.dominiomarcelino.com/
http://www.fentowines.com/es
http://www.fincamillara.com/
http://www.novatoural.es/
http://www.rectoraldegundivos.es/
http://www.regoa.es/
http://www.virxendosremedios.es/
http://www.xn--saias-qta.com/
https://abasolleira.com/
https://adegabarbado.wordpress.com/
https://adegacruceiro.es/web/
https://almadasdonas.com/
https://alvaredoshobbs.com/es/inicio/
https://amandeprado.com/
https://atriumvitis.com/
https://bodegaedv.com/
https://bodegapincelo.es/
https://bodegaspetron.es/
https://ciudaseis.com/
https://fazendapradio.com/es/
https://guimaro.es/
https://oscipreses.com/
https://pombares.com/
https://pontedaboga.es/
https://quinta89.negocio.site/
https://rectoraldeamandi.com/
https://reginaviarum.es/
https://ronseldosil.com/es/
https://www.adegacachin.com/
https://www.adegasmoure.com/
https://www.adegatear.com/
https://www.bodeganogueira.com/
https://www.casamoreiras.com/
https://www.chaodocouso.com/
https://www.condadodesequeiras.com/
https://www.donbernardino.com/
https://www.fincacuarta.es/
https://www.losadavinosdefinca.com/bodega

https://www.pazodelacuesta.com/
https://www.valdalenda.com/
https://www.viaromana.es/
http://bodegaslaflorida.com/
http://www.bodegasmartinon.com/
https://bodegasguiguan.com/
https://bodegasrubicon.com/
https://bodegavulcano.es/
https://elgrifo.com/
https://stratvs.com/
https://vinoslosperdomos.com/
https://www.fincafajardo.com/
https://www.lageria.com/
https://www.losbermejos.com/
https://www.vegadeyuco.com/

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
