# Peer review of "Managing Digital Presence in Wineries Practicing Heroic Agriculture: The Cases of Ribeira Sacra and Lanzarote (Spain)"

_agronomy, doi:10.3390/agronomy13030946_

Round 1
Reviewer 1 Report
dear authors
this paper is very interesting
please see the attached file

Author Response
Dear reviewer,
thank you for your consideration. They have helped us to improve our research. The most critical text changes are marked in red.
• All the bibliographic references have been corrected and adapted to the journal format.
• The term "Denominación de Origen" (DO) has been translated into Appellation of Origin using the acronym AO.
• The term Research Question has been corrected from Qn to RQn.
• It has been explained how the research fits within the contributions made in academic research and the novelties it brings.
• A paragraph has been added mentioning similar research and case studies that address the use of social networks in the field of wine tourism.
• Improvements have been made to the results.
• The changes suggested in points 7, 8, 9 and 10 have been addressed.
• The attributes of terroir wines, the sources used, and the references have been specified to provide further coherence to the research. The results lead to conclusions that refer to the research questions.
• Concerning heroic agriculture, the study, as explained, aims to establish a comparison between two production areas that are included in what is known as heroic viticulture. Given the peculiarities of these areas, the content analysis yields exciting results.
• A paragraph has been introduced in the discussion in which reference is made to similar studies and the significant differences that have been found.
Reviewer 2 Report
After the review of the study, some suggestions are provided:
1. Introduction: only a brief introduction of the two case studies is suggested in this section. The details concerning the two Spanish areas considered for this research can be reported in a dedicated section in Materials and Methods.
2. Scientific methods and methodological choices:
a. It is suggested to provide more details to the description of the study from Huertas (2016), which results to be crucial for the content analysis of this research.
b. It is suggested to deepen the information about the reasons for considering Instagram as a social network where you mainly focused your qualitative analysis (e.g. functions, aims, mode of uses).
c. It is suggested to provide more information about the software NVivo used for the study, particularly related to its functions for the qualitative analysis of pictures.
d. It is suggested to specify the whole list of the websites (or wineries) consulted in the text.
3. Methods section: to facilitate more fluent reading, it is recommended to merge the text on the analysis of tourist presence (research step 1) and the qualitative analysis of Instagram websites (research step 2). The description of the two steps (from line 250 to 264) alternates repeatedly.
4. Results should be presented in a more detailed way. For example, figure 9 needs a deeper description of the steps bringing to these results.
5. References and in-text citations:
a. It is strongly suggested to provide references in the tables and figures (see the description of Table 1; see line 293: to specify).
b. For in-text quotes, it is suggested to provide the n. page of the references quoted (see line 158).
c. Generally, it is suggested to review the in-text citation and the references list according to the journal's criteria.
6. General text and writing:
a. Acronyms: to allow a more fluent and accessible reading, it is strongly suggested to provide all the acronyms after the full description of them. It is suggested to do this on the first occasion they are treated in the text and thereafter to use only acronyms (see lines 18 and 19: it is not immediate to understand what a ‘DO’ is when no extended information about the term is provided; see line 108: CRDO). Moreover, it seems that the acronyms AO and DO are used interchangeably. To avoid confusion, it is suggested to specify possible connections/differences and uses the authors make of the two terms in the text.
b. Extensive editing of English language and style is strongly suggested (for example, see the description of Figures 6,8,9,10 to translate or line 306 to lighten).
c. Sectorial terms and phenomena: it is suggested to provide information about all the sectorial terms and phenomena important for this study (see, for example, line 35: it is suggested to define what a ‘geographical demarcation’ is briefly, while leaving out the details of phenomena that are not strictly addressed by the study (see, for example, line 144 and ‘certified routes’. This implies to intrigue the reader to question why and how routes are certified, thus distancing him/her from the purposes of the research. Hopefully, this would help to make the study accessible to the broader public and not only to the experts.
Author Response
Dear Reviewer:
Thanks you for your suggestions. They have helped us to improve the research. A native translator has revised the text to eliminate any inaccuracies. The most critical text changes are marsked in red.
• A description of the territories under study has been included in the section "Materials and Methods as suggested".
• The reference to the Paniagua-Rojano h Huertas study has been improved, and the importance of Instagram in the study has been explained, as he indicated.
• A paragraph has been included explaining in more depth how Nvivo was used.
• The websites of all the wineries under study have been included in Appendix A.
• The repetitions indicated in the methodology have been corrected, and an essential part of this section has been rewritten to improve its readability and clarify concepts.
• Clarifications have been introduced to improve the understanding of the results.
• Where necessary, references have been introduced in the tables and figures. Likewise, the pages in the references have been specified where necessary.
• All citations and references have been revised concerning the journal's standards.
• The terms that appeared in Spanish have been corrected and translated, and acronyms have been used appropriately.
• We have proceeded to explain some terms, such as Geographic Demarcation Dear reviewer, thank you for your comments.

Reviewer 3 Report
Interesting research.
However it would be important to further clarify the concept of Heroic Viticulture, considering an approach based on a historical and cultural view of viticulture, mainly in Europe (cases of Italy, France, Germany or Portugal).
It was felt the lack of bibliography concerning these aspects, that authors such as Hugh Johnson, Gilbert Garrier and others could provide. We enlighten this: even a managing co-creation paper such as this - and mainly when the results show, so clearly, the importance of cultural heritage in the wine destination image - should have a wider cultural and social approach.
An interview/inquiry to residents and tourists would be a richer way to research, more profoundly, these matters. Digital and social media aren't enough to cover identity topics, in our point of view, mainly after pandemic time (and the paper enphsizes this post-COVID orientation).
We wish all the best and congratulate the authorship.
Author Response
Dear reviewer, thank you for your comments.
We know the contributions of Hugh Johnson and Gilbert Garrier that have been collected by later studies such as Getz, Brown, Robinson or Sigala, among others. His cultural perception of wine is a contribution assumed by academics and experts.
Between lines 78 to 85, it defines what Heroic Agriculture means and refers to CERVIM, the study centre that houses many European areas dedicated to heroic Viticulture.
As described, the object of the investigation is to know the digital presence of the wineries, the communicative values present in the communication strategies and the way social media are used. Carrying out a study of cultural elements is beyond the scope of this investigation, although it would be interesting to propose a study along these lines.
In the case of conducting interviews with wineries, something similar occurs. It moves away from the proposed object of study, although it is still an interesting approach to consider.
Reviewer 4 Report
The presented work deals with the possibilities offered by social media to promote rural territory and wine production through wine tourism sustainably. The paper is well written. I do not see any omission of an important aspect of the problem.
I have one comment only to consider for the authors about methodology:
Might be helpful some more details about the survey employed. It would be useful to show the period of data collection and to quantify the media and documents that have been analyzed, how systematically the data collection took place, etc. I believe that the methodology should not raise questions for the reader.
Author Response
Thank you for your comments.
They have helped us to improve the research. A native translator has revised the text to eliminate any inaccuracies. The most critical text changes are marsked in red.
We have improved the wording of the methodology, where you can find a complete and detailed description of when and how the study sample was developed. An Appendix has been added in which the websites of the wineries studied are listed.